



# Quantifying erosion in a pre-Alpine catchment at high resolution with concentrations of cosmogenic $^{10}$Be, $^{26}$Al, and $^{14}$C

Chantal Schmidt[1,2], David Mair[1], Naki Akçar[1], Marcus Christl[3], Negar Haghipour[3], Christof Vockenhuber[3], Philip Gautschi[3], Brian McArdell[2], Fritz Schlunegger[1]

[1] Institute of Geology, University of Bern, Bern, 3012, Switzerland
[2] Swiss Federal Research Institute WSL, Birmensdorf, 8903, Switzerland
[3] Laboratory of Ion Beam Physics, ETH Zurich, Zürich, 8093, Switzerland

*Correspondence to*: Chantal Schmidt (chantal.schmidt@unibe.ch)

**Abstract**

Quantifying erosion across spatial and temporal scales is essential for assessing different controlling mechanisms and their contribution to long-term sediment production. However, the episodic supply of material through landsliding complicates quantifying the impact of the individual erosional mechanisms at the catchment scale. To address this, we combine the results of geomorphic mapping with measurements of cosmogenic $^{10}$Be, $^{26}$Al, and $^{14}$C concentrations in detrital quartz. The sediments were collected in a dense network of nested sub-catchments within the 12 km$^2$-large Gürbe basin that is situated at the northern margin of the Central European Alps of Switzerland. The goal is to quantify the denudation rates, disentangle the contributions of the different erosional mechanisms (landsliding versus overland flow erosion) to the sedimentary budget of the study basin, and to trace the sedimentary material from source to sink. In the Gürbe basin, spatial erosion patterns derived from $^{10}$Be and $^{26}$Al concentrations indicate two distinct zones: headwater zone with moderately steep hillslopes dominated by overland flow erosion, with high nuclide concentrations and low denudation rates (~ 0.1 mm/yr), and a steeper lower zone shaped by deep-seated landslides, where lower concentrations correspond to higher denudation rates (up to 0.3 mm/yr). In addition, $^{26}$Al/$^{10}$Be ratios in the upper zone align with the surface production ratio of these isotopes (6.75), which is consistent with sediment production through overland flow erosion. In the lower zone, higher $^{26}$Al/$^{10}$Be ratios of up to 8.8 point towards sediment contribution from greater depths, which characterises the landslide signal. The presence of a knickzone in the river channel at the border between the two zones points to the occurrence of a headward migrating erosional front and supports the interpretation that the basin is undergoing a long-term transient response to post-glacial topographic changes. In this context, erosion rates inferred from $^{10}$Be and $^{26}$Al isotopes are consistent, suggesting a near-steady, possibly self-organised sediment production regime over the past several thousand years. In such a regime, individual and stochastically operating landslides are aggregate over time in a specific region of higher erosion with a higher average denudation rate. Although in-situ $^{14}$C measurements were also conducted, the resulting concentrations show a non-conclusive pattern.





## 1 Introduction

In alpine environments stochastic processes such as landslides often drive sediment production and condition the occurrence

of debris flows  (Kober et al., 2012; Clapuyt et al., 2019). During periods of strong hillslope-channel coupling, the processes operating on the hillslopes deliver detrital material to the channel network, where sediment from various sources becomes mixed and transported downstream. As a consequence, the sediments at the outlet of an alpine catchment are a mixture of detrital material generated through a large variety of erosional mechanisms in different locations in the upstream basin. This makes it challenging to allocate the detrital material and to quantify how the different sub-catchments and erosional processes

have contributed to the overall sediment budget. This is particularly the case for those basins that are underlain by a homogenous bedrock lithology, which prevents the identification of different sediment sources using petrologic fingerprinting methods (e.g., Stutenbecker et al., 2018).  In such a context, in-situ $^{10}$Be has proven a useful tool to quantify the generation of sediment through erosion (Bierman and Steig, 1996; vonBlanckenburg, 2005) across a large range of catchment sizes – from small headwater basins ($\sim$1 km$^2$; Granger et al., 1996) to major river systems such as the Ganges and Amazon rivers (Wittmann

et al., 2009; Dingle et al., 2018;). In addition, $^{10}$Be-derived denudation rates have also been successfully applied to explore the controls of various parameters on surface erosion such as: topography, rock strength (DiBiase et al., 2010; Carr et al., 2023) environmental conditions (Reber et al., 2017), rock uplift as well as climatic variables including precipitation (Roda-Boluda et al., 2019), runoff and runoff variability (Savi et al., 2015). However, a successful $^{10}$Be-based assessment of basin-averaged denudation rates requires that the material at the sampling site is well mixed (Binnie et al., 2006), representing the contributions

from the various tributary basins according to the rates at which sediment has been generated in them. In catchments where sediment has been episodically supplied e.g. by landslides, denudation rate estimates may be biased towards the impact of a specific sediment source (Bierman and Steig, 1996; Savi et al., 2014), particularly if samples are collected in small basins (Yanites et al., 2009; Marc et al., 2019). Accordingly, erosion rate estimates for basins where the sediment production has largely been controlled by landslides requires a sampling strategy where the corresponding upstream size of the basin increases

with landslide area if the goal is to capture a stable long-term erosion rate signal (Niemi et al., 2005; West et al., 2014). This is also the main reason why few studies have targeted small catchments with stochastic sediment delivery (Niemi et al., 2005; Kober et al., 2012). Nonetheless, recent work (DiBiase, 2018) has demonstrated that landslides primarily introduce some scatter, but not a strong bias into erosion rate estimates. Furthermore, the use of paired cosmogenic isotopes with different half-lives, such as $^{10}$Be – $^{26}$Al (Wittmann and vonBlanckenburg, 2009; Wittmann et al., 2011; Hippe et al., 2012) or $^{10}$Be – $^{14}$C

(Slosson et al., 2022; Skov et al., 2019; Hippe et al., 2019; Kober et al., 2012) have enabled to reconstruct the occurrence of sediment storage in the source-to-sink sedimentary cascade. They also improved our understanding about the importance of transient erosional effects on the generation of the cosmogenic signals in fluvial material (e.g., Hippe et al., 2012).

Here, we use information offered by concentrations of cosmogenic $^{10}$Be, $^{26}$Al, and $^{14}$C in riverine quartz, which we combine with the results of geomorphic mapping. The goal is to (i) trace the origin of the sediments, (ii) document the influence of

landsliding on the long-term sediment fluxes, and to explore the scale-dependency – in space and time – of the resulting





cosmogenic signals. In contrast to most previous studies, we particularly target small basins to identify the impact of landslides on the generation of cosmogenic signals. To this end, we focus our work on the Gürbe basin situated at the northern margin of the European Alps. Erosion in this basin has been largely controlled by large variety of erosional processes including sediment supply through deep-seated landslides (do Prado et al., 2024), thus making this basin an ideal target for our goals. We thus conduct a dense sampling program in this catchment and combine the results of the three cosmogenic isotopes ([10]Be, [26]Al, and [14]C) to allocate the origin of the material, quantify the scaling – both in space and time – of the erosional processes operating in the study basin, and to trace the clastic material from the source to the sink.

## 2 Local setting

The study area, the 12 km[2] Gürbe catchment, is situated at the northern margin of the Swiss Alps (Fig. 1a). The Gürbe River, with a main channel approximately 8 km long, originates at an elevation of approximately 1800 m a.s.l. There the landscape is characterised by steep cliffs of Mesozoic limestones that are part of the Penninic Klippen belt (Jäckle, 2013) (Fig. 1b). These units are partially covered by a several-meter-thick layer of glacial deposits (i.e. till Swisstopo, 2024a). The orientation of the corresponding moraine ridges suggest deposition by small, local glaciers during the Last Glacial Maximum (LGM) ca. 20'000 years ago (Bini et al., 2009; Ivy-Ochs et al., 2022). The headwater area of the Gürbe catchment hosts a second main tributary, the Schwändligraben River (Fig. 1a), which originates within Cretaceous to Eocene Gurnigel-Flysch units comprised of alternating layers of marls, sandstones, polymictic conglomerates and mudstones (Winkler, 1984). The landscape in the source area of this tributary is characterised by a swampy terrain and ancient deep-seated gravitational slope failures. At approximately 1200 m a.s.l., a knickzone that corresponds to the highest glacial deposits of the LGM Aare-glacier in this region (Bini et al., 2009) separates the landscape in an upper and a lower zone. At this knickzone, the longitudinal profile of the Gürbe River steepens from originally 6.5° to 9.3° (Fig. 1c). Similarly to the region upstream of the knickzone, the bedrock in the lower part of the Gürbe basin is predominantly composed alternated sandstones and mudstones that either occur in the Gurnigel-Flysch unit (Swisstopo, 2024a) or in the Lower Marine Molasse (Diem, 1986). In this lower part of the Gürbe basin, the hillslopes are between 20° and 25° steep and covered by a dense forest made up of spruce.

Five areas prone for landsliding have been identified in the lower part of the Gürbe basin, where three of them are located on the SE side and two on the NW side of the valley (Zimmermann et al., 2016). These landslides are known for their re-current activities during the past decades either experiencing a slow, continuous movement or periodic reactivations. The dynamics of these landslides are characterised by short episodes of accelerated slip in the range of several meters to tens of meters per day, followed by period during which the landslides have been stable (Zimmermann et al., 2016). In locations where the toes of the deep-seated landslides reach the Gürbe channel, the surfaces of these hillslope failures are reworked by secondary shallow-seated landslides, marking fresh scars in the topography. Also, in this lower part of the catchment, the Gürbe River has been actively incising into bedrock, thereby re-activating most of these landslides and resulting in a relatively high mean annual sediment discharge of c. 10'000 to 20'000 m[3] at the downstream end of the Gürbe basin (Salvisberg, 2017). This was the main



reason why approximately 140 check dams have been built during the past century to stabilise the streambed, reduce the gradient, and thereby regulate the transport of bedload (Salvisberg, 2017; do Prado et al., 2024). At the downstream end of the
lower section, the Gürbe channel transitions into the deposition zone, forming an approximately 4 km$^2$ alluvial fan with a distinct apex situated at 800 m a.s.l. On this fan and farther downstream, the Gürbe River flows in an artificial channel that is stabilised by check dams and flood protection dikes. After passing through an artificial deposition area, the river enters the Gürbe valley floodplain, where the stream flows in a confined channel until merging with the Aare River approximately 20 km downstream.

The runoff conditions of the Gürbe River are characteristic for a pre-alpine environment, exhibiting a nivo-pluvial discharge regime (Salvisberg, 2017; Jäckle, 2013). Between 1981 to 2010 the discharge of the Gürbe River has been continuously measured at the Burgistein gauging station that is situated ca. 5 km downstream of the Grübe fan (Fig. 1a). During this period, the mean annual precipitation rates have ranged from approximately 1100 mm/year in the alluvial fan area to nearly 2000 mm/year in the headwaters of the catchment (Frei et al., 2018; based on MeteoSwiss, 2014). Due to the low water storage
capacity of the soils, including the soils in the Klippen Belt and the regolith cover of the highly saturated, low-permeability Gurnigel Flysch and Lower Marine Molasse units, the catchment rapidly responds to high rainfall rates, resulting in peak floods with short durations (Ramirez et al., 2022). The gauging records indeed show that intense summer thunderstorms, with rainfall intensities up to 30 mm/h, tend to generate large discharge events. The highest recorded water discharge of 84 m$^3$/s occurred on the 29$^{th}$ of July 1990, (Ramirez et al., 2022; do Prado et al., 2024). In contrast, the mean annual discharge has been
approximately 1.3 m$^3$/s during the survey period. Such high discharge variabilities emphasise the torrential character of the Gürbe River (Ramirez et al., 2022; Salvisberg, 2022).

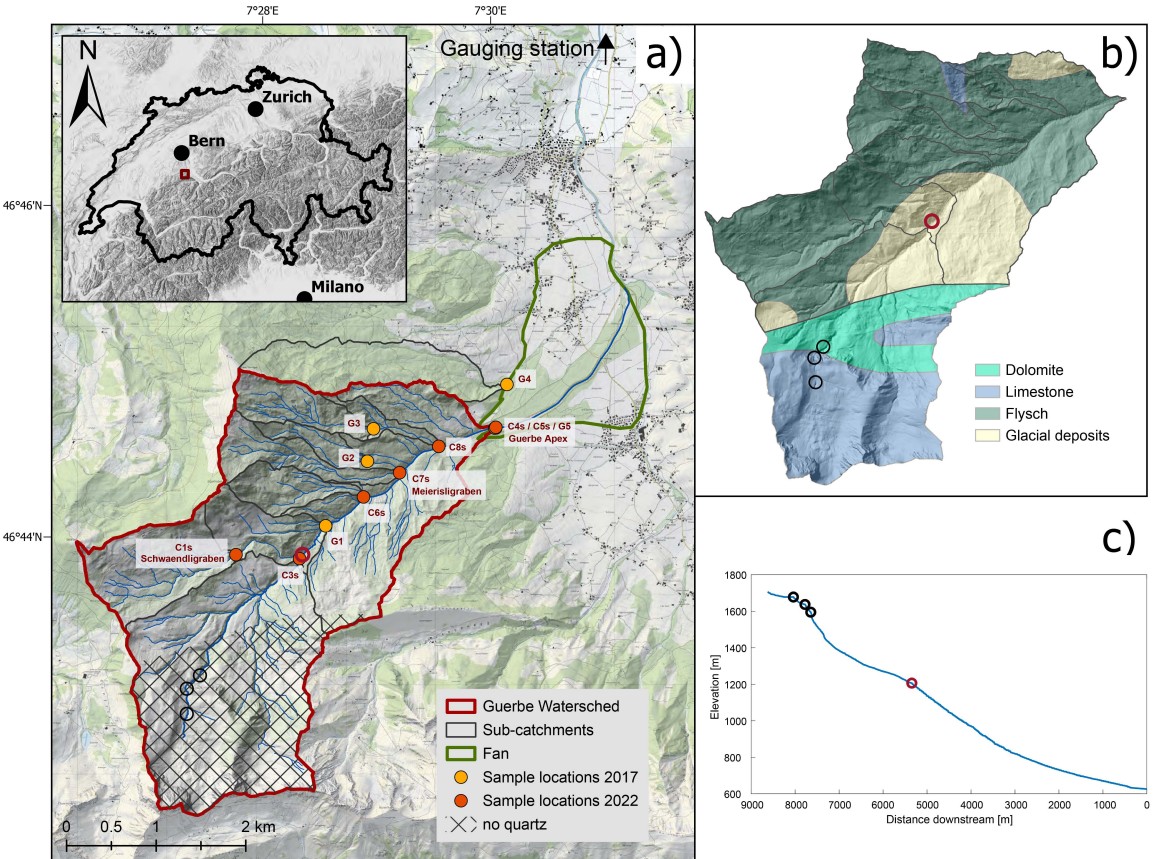

**Figure 1: Overview of the Gürbe watershed.** (a) Study site showing the locations where cosmogenic samples were taken during the two sampling periods in 2017 (Latif, 2019) and 2020 and illustrating the corresponding sub-catchments. The no-quartz area, which is due to the lithology, was excluded from the calculations of the denudation rates and the morphometric properties; (b) simplified lithological map, and (c) longitudinal profile of the Gürbe main channel. Note that the occurrence of knickpoints is indicated by circles. The knickzone, situated at c. 1200 m a.s.l., corresponds to the height of the LGM glacier in this region (red circle). It marks the transition from the upper to the lower part of the catchment. Further knickzones in the upper part (black circles) are conditioned by differences in the lithologic architecture of the bedrock. Digital elevation model (DEM) from the Federal Office of Topography swisstopo (Swisstopo, 2024d) and overview map relief from (Bundesamt für Landestopografie swisstopo et al., 2023).

## 3 Methods

Following the scope of the paper, we integrated geomorphic and geologic data with cosmogenic nuclide-derived estimations of catchment-averaged denudation rates to yield a detailed picture on the origin of the sedimentary material and the source-to-sink sedimentary cascade of the clastic detritus. We used the swissAlti3D high-resolution digital elevation model (DEM; Swisstopo, 2024d) with a spatial resolution of 2 m as a basis, which we then re-sampled to a 5 m-DEM for further scaling analysis. We justify this re-sampling to a lower resolution because a grid with a lower resolution prevents the introduction of




artefacts upon calculating the flow paths. Additionally we corrected the DEM in the Meierisligraben area to account for the deflection of the corresponding channel, a feature already documented in the swisstopo map 1:10'000 (Swiss Map Raster 10; Swisstopo, 2024c). The correction was performed by digitizing a new DEM using the contour lines delineating the affected area and merging the modified area with the resampled 5 m-DEM. The finalised 5 m-DEM was then used for (i) the quantification of the morphometric properties of the study area contributing to the generation of cosmogenic isotopes (area that supplies quartz), see section 3.1, (ii) the establishment of a geomorphic map illustrating the erosional processes within the Gürbe basin, and (iii) the calculation of the catchment-averaged denudation rates from the measured concentrations of the cosmogenic isotopes $^{10}$Be, $^{26}$Al and $^{14}$C.

## 3.1 Morphometric characterisation

We delineated the different sub-catchment areas in ArcGISPro (v3.1) using the 5 m-DEM as a basis (see previous section). In addition, we computed the sediment connectivity index with the algorithm developed (Cavalli et al., 2013) and implemented as an ArcGIS toolbox in ArcMap (v.10.3.1). A compilation of geologic data includes the collection of information on quartz-bearing lithologies, the occurrence of glacial deposits and the extent of the LGM. We gathered such data from the three map datasets including i) GeoCover (Swisstopo, 2024a) based on Heinz et al. (2023) and Tercier and Bieri (1961), ii) Geological Map of Switzerland 1:500'000, and iii) Map of Switzerland during the Last Glacial Maximum (LGM) 1:500'000 (GeoMaps 500 Pixel; Swisstopo, 2024b). Based on this data we grouped the stratigraphic units into quartz-bearing and non-quartz-bearing (Fig. 1a). As a subsequent step, we corrected the catchment shapes for the occurrence of quartz minerals in bedrock and glacial deposits, thereby clipping the steep limestone lithologies in the upper part of the catchment where quartz is absent (Figs. 1a and 1b). For the remaining sub-catchments, which contribute to the cosmogenic signals in the channel network because they contain quartz grains, we computed a range of routinely used topographic metrics including slope gradient, terrain ruggedness and channel steepness (e.g., Delunel et al., 2020). Slope gradient and terrain roughness were estimated using tools implemented in ArcGIS (v3.1) pro and the terrain ruggedness index (TRI) by Riley et al. (1999).

The extraction of river profiles, the calculation of the normalised channel steepness index ($k_{sn}$) and the mapping of knickpoints was accomplished with the corresponding tools of Topotoolbox (v2.2, Schwanghart and Scherler, 2014). For these calculations, we considered stream segments that have an upstream drainage area >2000 pixels, and that contribute to the generation of detrital quartz grains. In addition, $k_{sn}$ values were calculated on 125 m-long channel segments, thereby employing a $k_{sn}$-radius of 500 m. This corresponds to a threshold area of 1000 m$^2$ and a minimum upstream accumulation area of 0.01 km$^2$. As reference concavity we used the default value of $\theta = 0.45$ (Schwanghart and Scherler, 2014).

## 3.2 Mapping

It has been shown that landsliding, incision and overland flow erosion are the most important erosional mechanisms contributing to the generation of sediment in a pre-Alpine catchment (Battista et al., 2020). Accordingly, we created an inventory of such processes and potential sediment sources using a variety of methods and data sources. We reconstructed the



spatial distribution of landslides through (i) a compilation of already published maps, (ii) an analysis of information available
from historical orthophotos, and (iii) observations in the field. Here, we used the landslide inventory from the Natural Hazards
Event Catalogue referred to as Naturereigniskataster (NGKAT, 2024) as a basis, which is available from the 'Geoportal des
Kantons Bern'. We updated this database with information that we retrieved upon analysing historical orthophotos. In this
context, the black and white orthophotomosaic images cover the time interval between 1946 and 1998 (SWISSIMAGE HIST;
Swisstopo, 2024f), whereas the time span between 1998 to the present are represented by the three consecutive
orthophotomosaics. These latter images have a better spatial resolution than the previous ones (SWISSIMAGE 10 cm;
Swisstopo, 2024e). We thus used this dataset as a basis to reconstruct the occurrence of topographic changes in the Gürbe
catchment over multiple decades. We complemented this dataset with observations on the hillshade DEM, and the maps
displaying the slopes and terrain roughness index (TRI) to complete the inventory of landslide occurrence in the study area.
We additionally used the given information to distinguish between shallow landslides and thus surficial processes, and deep-
seated movements. Shallow landslides, typically covering areas of a few square meters, were identified based on the occurrence
of (i) erosion scars resulting in surfaces that are free of vegetation, (ii) small scale depressions in the head of the inferred
landslides ('Nackentälchen' according to Haldimann et al., 2017), and (iii) regions characterised by a down-slope increase of
the hillslope angle because an escarpment, characterizing the head of a landslide, is typically steeper than the stable area farther
upslope (visible in the slope and TRI, see also Schlunegger and Garefalakis, 2023). In contrast, the deep-seated landslides that
cover larger areas are characterised by the occurrence of (i) potentially vegetation free areas, (ii) altered vegetation such as
sparse or smaller trees with a light colour on the satellite imageries, and (iii) zones of slip that are marked by sharp edges
separating a smooth topography above the escarpment edge from an uneven, bumpy terrain below it. Upon mapping landslides,
we paid special attention to recognise whether such sediment sources were connected with the channel network, thereby
contributing to the bulk sedimentary budget of the Gürbe River.
Incised areas are generally characterised by a V-shaped cross-sectional geometry (Battista et al., 2020). Accordingly, we
mapped the upper boundary of such a domain where the landscape is characterised by a distinct break-in-slope. There, the
landscape abruptly changes from a gentle morphology above this edge to a steep terrain farther downslope (e.g., Fig. 2C in
Cruz Nunes et al., 2015). The steep slopes ending in a channel forming such incisions usually expose the bedrock or the non-
consolidated material into which the incision has occurred. Similar to the approach of van den Berg et al. (2012) and Battista
et al. (2020), we used the aforementioned criteria to identify such incised areas on the DEM and in the field. Furthermore, we
mapped areas where sediment generation has been accomplished through overland flow erosion. In a landscape, such segments
are characterised by gentle slopes and smooth transitions between steep and flat areas. They can be readily identified in a
hillshade-DEM by their smooth surface texture, lacking evidence for the occurrence of bumps, hollows and scars (Battista et
al., 2020; van den Berg et al., 2012). Finally, we also mapped the occurrence of scree deposits, which occur locally at the base
of limestone cliffs. All information were assembled and synthesised in an ArcGIS environment to produce the geomorphic
map, illustrating the areas of landslide occurrence.





### 3.3 Catchment-wide denudation rates inferred from concentration of cosmogenic [10]Be, [26]Al and [14]C

#### 3.3.1 Sampling strategy and data compilation

During autumn 2022 and spring 2023 we collected a total of seven riverine sand samples at six locations for the analysis of
the cosmogenic nuclides [10]Be, [26]Al and [14]C. Each sample consisted of 2–4 kg of sand taken from the active channel bed. We sampled material at those sites where we anticipate capturing the different source signals that result from the various erosional mechanisms (overland flow erosion, incision, and landsliding, see section 3.2). Accordingly, the uppermost sample (C1s) is taken in the Schwändligraben River, which is the main upstream tributary (Fig. 1a). Mapping suggests that at this site the concentration of the cosmogenic isotopes will characterise the signal related to overland flow erosion, or alternatively to what
has been referred to as hillslope diffusion in the modelling literature (e.g., Tucker and Slingerland, 1997). A second sample (C3s) was collected in the Gürbe stream just upstream of the confluence with the Schwändligraben River. Similarly to sample C1s, the riverine material taken at site C3s is expected to record a cosmogenic signal related to overland flow erosion, yet it is expected to partially also record the contribution of material derived from the incised area (see section 3.2). We then sampled material from three tributaries draining the area with the deep-seated landslides on the NW orographic left side (samples C6s,
C7s and C8s). Finally, we collected two sand samples at the outlet of the catchment and thus at the Gürbe fan apex (C4s and C5s), where the stream starts to flow on the fan. Here, the goal is to characterise the cosmogenic signal representing the mixture of the supplied material from further upstream. We took one sand sample from the active channel and another one from a higher-elevation, presumably older gravel bar that is situated on the left lateral margin of the Gürbe channel. In addition, these two separate samples were collected to evaluate the consistency of the mixed sediment signal at the outlet of the catchment.
We also included [10]Be concentrations of 5 samples (G1 to G5) collected in 2017 and presented in Delunel et al. (2020) and Latif (2019) in our analysis (Fig. 1a). Site G1 is situated ca. 500 m downstream of the confluence between the Gürbe and Schwändligraben rivers and downstream of the incised area. Accordingly, we anticipate that the [10]Be concentration of the riverine sand from this site carries the erosional signal related to the combined effect of overland flow erosion (or hillslope diffusion) and fluvial incision (van den Berg et al., 2012; Battista et al., 2020). Sites G2 and G3 are located in tributary channels
on the orographic left side where sediment is derived from the landslides. Similarly to sites C6s to C8s, we expect the concentrations at these sites to characterise the cosmogenic signals related to a material supply from the deep-seated landslides. Site G5 is located on the Gürbe fan apex and is expected to capture the erosional signal of the entire Gürbe catchment. Finally, site G4 is situated in a neighbouring river, which collects the sedimentary material from a drainage basin that is perched on a deep-seated landslide adjacent to the Gürbe catchment farther to the North. We included the [10]Be concentration from site G4
in our study because surface erosion in this catchment is not conditioned by the processes of the Gürbe River, thus yielding an independent constrain on denudation occurring in an area where deep-seated landslides have shaped the landscape and contributed to the generation of detrital material.





### 3.3.1 Laboratory work

The sand samples were sieved to isolate the 250 – 400 µm grain size fraction. However, due to insufficient material in this size
range, the 400 µm – 2 mm sand fraction was subsequently crushed and re-sieved to obtain sufficient material with the desired
grain size. Further sample processing was performed according to the protocols originally reported in Akçar et al. (2017). The
– 400 µm fraction underwent magnetic separation as a preliminary step, followed by the removal of carbonate minerals
trough HCL and successive leaching treatments using hydrofluoric acid, phosphoric acid, and aqua regia to purify the quartz.
After measuring the total Al content, additional leaching was performed to further reduce impurities and lower the total Al
concentrations. Approximately 40 g of purified quartz was spiked with the $^9$Be carrier BL-SCH-3 and dissolved in concentrated
hydrofluoric acid. Be and Al were stepwise extracted from this solution using anion and cation exchange chromatography,
thereby following the protocol of Akçar et al. (2017). The extracted and precipitated Be and Al isotopes were oxidised and
pressed into targets for the subsequent measurements of the $^{10}$Be/$^9$Be and the $^{26}$Al/$^{27}$Al ratios with the accelerator mass
spectrometry (AMS) MILEA facility at the ETH Zürich (Maxeiner et al., 2019). The measured $^{10}$Be/$^9$Be ratios were normalised
with the ETH in-house standard S2007N (Kubik and Christl, 2010) and corrected using a full process blank ratio of 3.92 ±
1.07 x 10$^{-15}$. The measured $^{26}$Al/$^{27}$Al ratios were normalised to the ETH AMS standard ZAL02N (equivalent to KNSTD, Kubik
and Christl, 2010; Christl et al., 2013) and corrected for a small constant background rate. Total Al concentrations, which we
measured with the ICP-MS in the aliquot taken from each sample solution, are used to calculate $^{26}$Al concentrations from the
isotopic ratio.

For six samples (C1s, and C3s to C7s) three to four grams of the purified quartz were further processed at the ETH facility for
measuring the concentrations of the in-situ $^{14}$C (Lupker et al., 2019). The extraction of this isotope was performed following
the updated protocol of Hippe et al. (2013). Accordingly, we heated the cleaned quartz in a degassed platinum crucible under
ultra-high-purity oxygen to remove contaminants. The $^{14}$C isotopes were then extracted by heating the sample to 1650°C in
two cycles for 2 hours each, with an intermediate 30 minutes-long heating step at 1000°C. The extraction was performed under
static O$_2$ pressure in the oven. At the end of the extraction line, the purified CO$_2$ was trapped and flame sealed using LN. The
released gas was purified using cryogenic traps and a 550°C Cu wool oven, and the amount of pure CO$_2$ was measured
manometrically before being prepared for the subsequent AMS analysis. CO$_2$ samples were analysed using the MICADAS
200 kV AMS instrument with a gas ion source at the ETH/PSI, which enables the measurement of small carbon samples (2–
100 µg C) without requiring graphitization (Synal et al., 2007). The reduction of the AMS data followed the protocols outlined
by Hippe et al. (2013) and Hippe and Lifton (2014).

### 3.3.3 Calculation of denudation rates

We calculated catchment wide denudation rates using the online erosion rate tool formerly known as the CRONUS-Earth
calculator (v3, Balco et al., 2008), employing a time independent production scaling (Lal, 1991; Stone, 2000). For the bedrock
density, we utilised the widely used value of 2.65 g/cm$^3$. We corrected the scaling calculations for quartz-free regions by using



centerpoints and mean catchment elevation derived for only the area that is made up of quartz-bearing lithologies (see above). Following recommendations of DiBiase (2018), no topographic shielding correction was applied. As shielding due to temporary snow cover leads to a decrease of in-situ cosmogenic nuclide production rates for $^{26}$Al and $^{10}$Be (e.g., Delunel et al., 2014), we applied a snow cover shielding factor using the empirical approach of Delunel et al. (2020), which itself is based on Jonas et al. (2009). Therein, snow shielding factors were determined as a function of the catchment's mean elevation.

The erosion rates inferred from cosmogenic nuclides are frequently used to estimate the time needed to erode the uppermost 60 cm of material under the assumption of steady state denudation, which is referred to as integration time (Bierman and Steig, 1996; Granger et al., 1996). However, due to landslide perturbation we instead calculated the minimum exposure age as a proxy for the integration time. We thus calculated such time spans for all our nuclide-derived denudation rates, using the same online calculator under the assumption of no erosion and no inheritance.

**3.4 Whole-rock geochemical analysis**

In order to determine how the supply of sediment from various sources influence the overall sedimentary budget of the Gürbe catchment and the composition of the material in the channel network, we proceeded in a similar way as Glaus et al. (2019), Stutenbecker et al. (2018) and Da Silva Guimarães et al. (2021), measuring the whole rock composition of seven samples (C1s, C3s – C8s). The whole rock geochemical analysis was performed by Actlabs in Canada on sample material bevor separating

the magnetic constituents (see section 3.3.1 above) with grain size < 250µm. Elements and major oxides were measured in lithium borate fusion glasses through Inductively Coupled Plasma Mass Spectrometry (ICP-MS). The analytical package included the following major element oxides: $SiO_2$, $Al_2O_3$, $Fe_2O_3$, MnO, MgO, CaO, $Na_2O$, $K_2O$, $TiO_2$, $P_2O_5$ as well as the trace elements Ba, Sr, Y, Sc, Zr, Be, V. Additional elements Ag, Cd, Cu, Ni, Pb, Zn, S were measured based on multiacid digestion and TD-ICP. Correction was done by the loss of ignition (LOI). We performed principal component analysis (PCA)

with the scikit learn package in Python (Pedregosa et al., 2011) on the geochemical data with the aim to identify different source lithologies and to fingerprint potentially different sediment composition from the individual sub-catchments.





## 4 Results

### 4.1 Mapping, inferred erosional mechanisms and morphometric characterisation

The mapping results allow us to distinguish between three main zones within the Gürbe catchment, which are the upper zone, the lower zone and the fan area (Figs. 1a, 2a and 2b). The upper zone has predominantly been affected by small, shallow landslides (Fig. 2a). The resulting scars are visible in the historic and current ortho images. The displaced material has in most cases no obvious connection to the drainage network. Additionally, a few scree deposits exist at the base of limestone cliffs

(Fig. 2a); however, as they are not linked to the drainage network, they do not contribute to the sediment budget of the river network. Overall, the landscape was smoothened by glacial and periglacial processes during the LGM, while the channel network is distinctly defined by its incised nature (Fig. 2a), creating sharp, yet locally constrained escarpments between the hillslopes and the riverbanks. This upper zone is also characterised by a generally low connectivity between the hillslopes and the channel network (Fig. 2c). Except for the spatially disconnected cliffs, the hillslopes are generally gentle to moderately

steep, displaying a mean hillslope angle of 19.2°, and the channel network is characterised by generally low mean normalised steepness values ranging between 26.5 $m^{-0.9}$ and 30.0 $m^{-0.9}$ for the two sub-catchments.

In the lower zone, numerous mid- to deep-seated landslides are observed (Fig. 2a). The landscape in this area is characterised by an absence of stable, deeply incised channels, with a channel network instead appearing less well developed. Apparently, the channels adjacent and perched on these landslides have been deflected multiple times during major landslide events (e.g.,

Simpson and Schlunegger, 2003). In comparison to the upper part, the connectivity between hillslopes and the channel network is higher (Fig. 2c), the hillslopes are generally steeper (mean average hillslope angle of 21.8°), and the tributary and as well as the trunk channels have generally higher steepness values, ranging from 32.4 $m^{-0.9}$ to 43.4 $m^{-0.9}$.

The boundary between these two zones is marked by a V-shaped erosional scarp indicating the occurrence of incision. This also aligns with the occurrence of the major knickzone along the Gürbe main channel, which is situated at around 1200 m a.s.l

and marks the transition reach from the upper to the lower zone (Figs. 1c and 2a). This knickzone reach is readily visible by the change in the steepness values of the channel network, which are higher along the incised reach than upstream and downstream of it.





**Figure 2: Geomorphic map of the Gürbe catchment.** (a) Mapping indicates two distinct geomorphic zones: 1) the southern, upper part of the catchment is characterised by incised channels, limited scree deposits, and shallow landslides; 2) the northern, lower part is a dissected and topographically complex landscape marked by multiple medium- to deep-seated landslides. (b) Delineation of three zones: the upper zone (blue), the lower zone (orange), and the sampling locations at the fan apex (red), which represents the site where sediment is supplied to the depositional area. (c) Index of connectivity with respect to the fan apex. The upper zone exhibits a low connectivity between hillslopes and the channel network, while the connectivity increases markedly in the lower zone, reflecting that the pathway of sediment transfer – relative to the fan apex – is more direct for sediment originating in the lower part than in the headwater region. Digital elevation model (DEM) from the Federal Office of Topography swisstopo (Swisstopo, 2024d).

## 4.2 Cosmogenic nuclide concentrations

The $^{10}$Be concentrations in all 12 samples range from the lowest value of $2.50 \pm 0.10 \times 10^4$ at/g (resp. $2.19 \pm 0.13 \times 10^4$ at/g) measured in the sample G4 to the highest value of $8.67 \pm 0.24 \times 10^4$ at/g measured in the uppermost sample C1 (Table 1, S2). A comparison of the values across the catchment show a distinct pattern across the study area. In the upper zone, characterised by sites C1s and C3s (Fig. 1a), the concentrations of in-situ $^{10}$Be are up to three times higher than those measured at the outlet of the Gürbe catchment (sites C4s, C5s and G5, Fig. 1a). This pattern suggests a significant supply of sediment with low $^{10}$Be concentrations to the Gürbe channel downstream of sites C1s and C3s. The tributaries, characterised by samples taken at sites





C6s–C8s, G2 and G3 (Fig. 1a), contribute sediment with concentrations at or below those recorded at the outlet (Table 1),

reinforcing their role as sources of material with lower concentrations. Additionally, among the three samples taken at the apex of the Gürbe fan, the two samples collected at different times (C4s & G5) show nearly identical $^{10}$Be concentrations (3.23 ± 0.21 & 3.27 ± 0.31 × 10$^5$ at/g). Similarly, also the third sample C5s, taken during the same survey as sample C4s but from a slightly different location within the active channel bed, displays a $^{10}$Be concentration that is nearly the same (within uncertainties) as the concentrations measured for the two samples from the same location.

The measured $^{26}$Al concentrations show a similar pattern to those of $^{10}$Be across the catchment. The lowest concentration (1.94 ± 0.01 × 10$^5$ at/g) was recorded in the same tributary as the lowest $^{10}$Be value, while the highest concentration (6.42 ± 0.24 × 10$^5$ at/g) was measured in the same uppermost sample (Table 1, S3). At the Gürbe fan apex, the two samples collected at different locations show no statistically significant differences in their concentrations (2.41 ± 0.28 × 10$^5$ at/g for sample C4s versus 2.27 ± 0.22 × 10$^5$ at/g for sample C5s).

The $^{14}$C concentrations of the six samples (C1s, C3s to C7s) range from 3.47 ± 0.10 × 10$^4$ at/g to 11.50 ± 0.23 × 10$^4$ at/g (Table 1, S5). While the lowest and highest concentrations are consistently recorded in the same two samples (C1s for the lowest and C7s for the highest concentration), the in-situ $^{14}$C concentrations of the other samples display different spatial patterns than the other nuclides. Additionally, the concentrations measured in the two samples at the fan apex differ from each other.

The $^{26}$Al/$^{10}$Be ratios range from 6.27 and 9.54 and, within a 2σ uncertainty, while the highest ratios of >7.76 are observed in

three tributaries. Ratios of $^{10}$Be/$^{14}$C range between 0.4 and 0.75.

**Table 1**: Measured $^{10}$Be, $^{26}$Al and $^{14}$C concentrations for all samples. All uncertainties given are 2σ and include the statistical uncertainties of the AMS measurement, the blank error (for $^{10}$Be) and ICP-MS measurement uncertainty (for $^{26}$Al).

| Sample | sampling location | | | $^{10}$Be | | | $^{26}$Al | | | $^{14}$C | | | $^{26}$Al/$^{10}$Be | | | $^{10}$Be/$^{14}$C | | |
|---|---|---|---|---|---|---|---|---|---|---|---|---|---|---|---|---|---|---|
| | Lat. | Long. | Alt. | Concentration | | | Concentration | | | Concentration | | | | | | | | |
| | [°] | [°] | [m. a.s.l.] | [10$^4$ at/g] | | | [10$^4$ at/g] | | | [10$^4$ at/g] | | | | | | | | |
| C1s | 46.73025 | 7.46171 | 1359 | 8.7 | ± | 0.5 | 64.2 | ± | 4.9 | 11.5 | ± | 0.5 | 7.4 | ± | 0.7 | 0.8 | ± | 0.1 |
| C3s | 46.72989 | 7.47103 | 1214.1 | 4.8 | ± | 0.3 | 34.8 | ± | 2.5 | 6.9 | ± | 0.6 | 7.2 | ± | 0.7 | 0.7 | ± | 0.1 |
| C4s | 46.7431 | 7.4998 | 765.6 | 3.2 | ± | 0.2 | 24.1 | ± | 2.8 | 7.1 | ± | 0.3 | 7.4 | ± | 1.0 | 0.5 | ± | 0.0 |
| C5s | 46.74296 | 7.49982 | 765.6 | 3.6 | ± | 0.2 | 22.6 | ± | 2.2 | 5.1 | ± | 0.4 | 6.3 | ± | 0.7 | 0.7 | ± | 0.1 |
| C6s | 46.73607 | 7.4804 | 1022.1 | 2.9 | ± | 0.2 | 25.7 | ± | 3.3 | 7.3 | ± | 0.3 | 8.8 | ± | 1.3 | 0.4 | ± | 0.0 |
| C7s | 46.73847 | 7.48562 | 927.2 | 2.5 | ± | 0.2 | 19.4 | ± | 2.2 | 3.5 | ± | 0.2 | 7.8 | ± | 1.1 | 0.7 | ± | 0.1 |
| C8s | 46.74119 | 7.49132 | 842.8 | 3.2 | ± | 0.2 | 30.6 | ± | 3.2 | | | | 9.5 | ± | 1.2 | | | |
| G1 | 46.73314 | 7.474846 | 1126.3 | 3.4 | ± | 0.3 | | | | | | | | | | | | |
| G2 | 46.739704 | 7.481027 | 1042.3 | 3.2 | ± | 0.3 | | | | | | | | | | | | |
| G3 | 46.742879 | 7.481919 | 1025.1 | 3.1 | ± | 0.3 | | | | | | | | | | | | |
| G4 | 46.747332 | 7.501382 | 724.3 | 2.2 | ± | 0.3 | | | | | | | | | | | | |
| G5 | 46.743069 | 7.499663 | 766.2 | 3.3 | ± | 0.3 | | | | | | | | | | | | |





### 4.3 Denudation rates

The erosion rates calculated for the three different nuclides are presented in Figure 3 (full data available in Table S6 in the
Supplement) against the upstream distance of the sampling location relative to the Gürbe fan apex. In general, the $^{10}$Be-derived
erosion rates range from 0.1 mm/yr to 0.3 mm/yr in one of the tributaries. Upstream of the knickzone, the $^{10}$Be-based erosion
rates are below 0.2 mm/yr, while downstream of the knickzone, the erosion rates increase to values between 0.2 mm/yr and
0.3 mm/yr. A comparison between $^{10}$Be- and $^{26}$Al-derived erosion rates shows that the rates are the same within uncertainties
in the upper catchment, but they differ in the tributaries downstream of the knickzone and at the fan apex. Specifically, $^{26}$Al-
derived erosion rates are higher than the corresponding $^{10}$Be-derived rates at the Gürbe fan apex, whereas in the tributaries
they are lower. The $^{14}$C-derived erosion rates range from 0.2 mm/yr to 1.0 mm/yr. They are thus up two to three times higher
than those derived from the other longer-lived nuclides. The differences $\Delta\varepsilon$ between the calculated denudation rates (Fig. 3)
range from 0.005 mm/yr up to 0.06 mm/yr ($\Delta\varepsilon$ between $^{26}$Al- and $^{10}$Be-based rates), and from 0.08 mm/yr up to 0.64 mm/yr
($\Delta\varepsilon$ between $^{14}$C- and $^{10}$Be-based rates).

The denudation rates presented above can be used to estimate the timespan over which the cosmogenic data have integrated
the erosional processes, which in our case, is approximately 3'000 to 6'000 years. More specifically, in a zero-erosion scenario,
the minimum duration of exposure required to accumulate the measured $^{10}$Be and $^{26}$Al concentrations ranges from 2'000 to
6'000 years, while for $^{14}$C concentrations it ranges from 800 to 3'000 years.

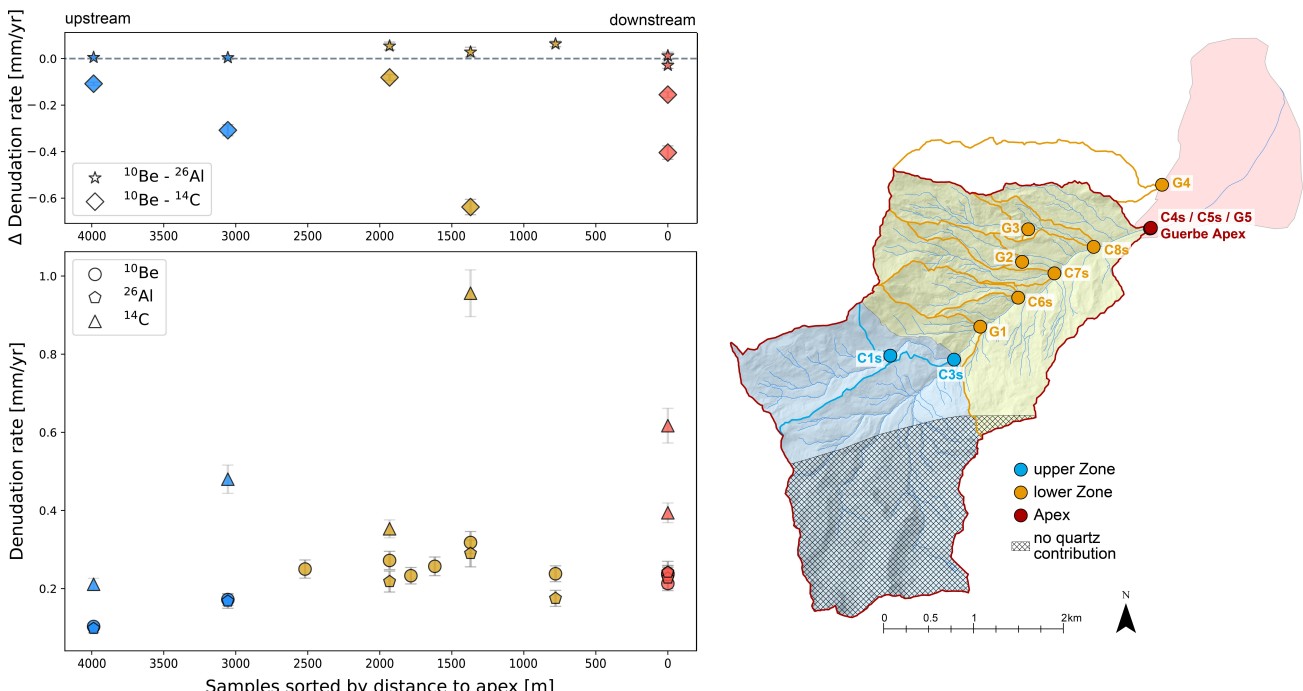

**Figure 3: Denudation rates calculated based on the measured $^{10}$Be, $^{26}$Al and $^{14}$C concentrations.** Each colour indicates the zone the
sample represents, with blue characterizing the upper zone, yellow the lower zone, and red the Gürbe fan apex. Digital elevation model
(DEM) from the Federal Office of Topography swisstopo (Swisstopo, 2024d).





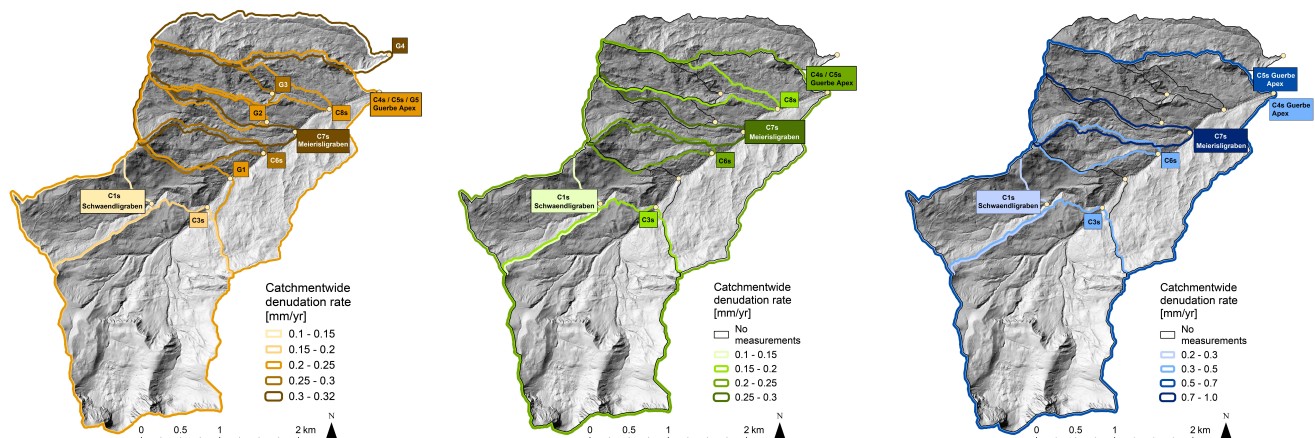

**Figure 4: Calculated denudation rates based on the ¹⁰Be (left), ²⁶Al (middle) and ¹⁴C (right) concentrations.** The lighter colours illustrate the highest concentrations (measured in the sample C1s representing the Schwändligraben sub-catchment in the upper zone). The lowest nuclide concentrations were measured in the Sample C7s, representing the Meierisligraben tributary in the lower zone. Digital elevation model (DEM) from the Federal Office of Topography swisstopo (Swisstopo, 2024d).

### 4.4 Geochemical sediment composition

The analysed samples showed variable $SiO_2$ content (33 -58 %-wt) and $Al_2O_3$ (5 – 12 %-wt), which are anticorrelated with CaO concentrations (8% to 29%; Table S6). Here, samples from catchments with limestone-bearing bedrock (e.g. C3s; Fig.2), show highest concentrations of CaO and lowest of $SiO_2$ and $Al_2O_3$ (Table S6), leading to a downstream decrease of CaO concentrations with a downstream decreasing relative proportion of limestone lithologies. All other major oxides did vary significantly between individual samples (Table S6). Similarly, trace elements concentrations did not vary strongly, with the main trace elements being Sr (325 ppm to 717 ppm), Ba (140 ppm to 310 ppm) and Zr (90 ppm to 220 ppm). Aside from the downstream decrease of limestone content in favour of $SiO_2$- and $Al_2O_3$-rich litharenites, no systematic variation of geochemical composition is visible in both major oxides and measured trace elements (Supplementary Fig. S1).



## 5 Discussion

The concentrations of the cosmogenic isotopes $^{10}$Be and $^{26}$Al and the resulting erosion rates show distinct differences between the upper, gently sloping region and the lower, steeper zone of the catchment (Figs. 3 and 4). We first discuss the implications of this pattern and particularly explore how the cosmogenic signals within the Gürbe basin change in the downstream direction as sediments derived from landslides impact the cosmogenic signal towards the catchment's outlet near the fan apex (section 5.1). Next, we combine the $^{10}$Be, $^{26}$Al, and $^{14}$C datasets to investigate the erosional dynamics across different temporal scales.

By comparing the signals preserved by these isotopes, we assess whether the cosmogenic nuclide concentrations reflect the occurrence of steady erosion over varying timescales, or if they record the effects of inheritance, burial, or transient perturbations over the same timescales (sections 5.2 and 5.3). As a next aspect, we discuss (i) how the cosmogenic nuclide-based erosion rates relate to the landscape's architecture by comparing them with the mapping results (section 5.4), and (ii) how this pattern has been conditioned by the glacial carving during the past glaciations (section 5.5). We end the discussion

with a notion that in drainage basins where the bedrock is too homogeneous to pinpoint the origin of the detrital material, terrestrial cosmogenic nuclides offer a viable tool for provenance tracing (section 5.6).

### 5.1 Downstream propagation and scale dependency of cosmogenic signals

   The $^{10}$Be and $^{26}$Al concentrations, and consequently the inferred denudation rates, record the occurrence of a variety of erosional mechanisms across the Gürbe basin. In the upper part of the catchment, high cosmogenic nuclide concentrations

correspond to low denudation rates. This contrasts with the lower nuclide concentrations and higher erosion rates inferred for the samples collected in the tributaries of the lower zone and at the fan apex (Fig. 5). This downstream decrease in concentrations of both nuclides suggests that the pattern of sediment generation has been stable over the erosional timescale recorded by them. Such an interpretation is corroborated by the same nuclide concentrations within uncertainties encountered in the three riverine samples at the fan apex. This is surprising because sediment supply through landsliding – in our case in

the lower zone – introduces a stochastic variability into the generation of sediment. Such a mechanism was already demonstrated for other Alpine torrents (Kober et al., 2012; Savi et al., 2014), where stochastic processes such as landslides and debris flows have resulted in episodic supply of sediment with low $^{10}$Be concentrations (Niemi et al., 2005; Kober et al., 2012). This has the potential to perturb the overall cosmogenic signal particularly in small catchments (Yanites et al., 2009; Marc et al., 2019), thereby (i) leading to variations in nuclide concentrations within riverine sediments collected from the same

channel bed (Binnie et al., 2006) and (ii) introducing scatter into the dataset (DiBiase et al., 2023). Given the small area (12 km$^2$) and the prevalence of recurrent landslides in the Gürbe sub-catchments, one might expect the sedimentary material at the Gürbe fan apex to record such variations. Yet our results indicate that the overall denudation signal has remained nearly stable at the fan apex (Fig. 5). This highlights an important scale-dependency in the erosional controls governing the generation of cosmogenic signals in the Gürbe basin. In particular, at smaller spatial scales (0.25 – 3.5 km$^2$), denudation rates reflect the

controls of local geomorphic and geologic conditions on erosion and material supply, such as repeated deep-seated landsliding





as in the case presented here. However, when these signals with a local origin are aggregated downstream towards the fan apex, they are recorded as a mixed, more stable signal that averages out the high and low concentrations generated in the individual sub-catchments. In the Gürbe basin, such mixing appears to occur at a spatial scale of less than 10 km$^2$. We thus infer – as this has already been mentioned by the many studies referred to in this paper in previous sections – that cosmogenic

nuclides remain a suitable tool for estimating long-term average erosion rates over thousands of years for landscapes eroding with rates between 0.1 and < 1 mmyr$^{-2}$ (vonBlanckenburg, 2005). This holds true even in catchments influenced by repeated stochastic processes – such as those documented for landslide processes in the Gürbe basin – provided that the sediment is sufficiently well mixed and that the corresponding cosmogenic nuclides are in an isotopic steady state (Clapuyt et al., 2019).

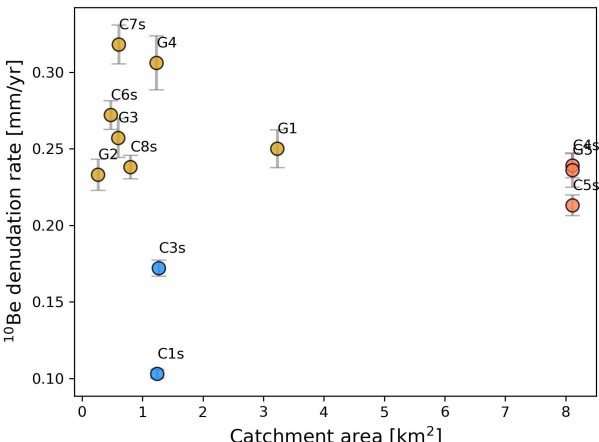


**Figure 5: Scale dependency of erosion rate estimates with concentrations of in-situ 10Be.** The variety of erosion rates determined for the scales of small local catchments (blue for the upper zone and yellow for the lower zone) is averaged out for the samples collected at the fan apex (red) where concentrations of in-situ $^{10}$Be characterise the mixed erosional signal of the entire catchment and thus for a larger scale.

## 5.2 Steadiness of the denudation signals across time scales

Concentrations of cosmogenic nuclides in riverine quartz have the potential to record the erosional history of a landscape, provided that (i) the nuclide production has occurred at a steady rate across the spatio-temporal scale over which they integrate erosion rates, and (ii) the nuclides are saturated (e.g., vonBlanckenburg, 2005). A potential deviation from these assumptions could be caused by a sedimentary legacy from past glaciations (Jautzy et al., 2024). This also concerns the Gürbe basin, as the study area has a history of erosion and material deposition by local and regional glaciers – most notably by the Aare glacier.

The coverage of the surface by glaciers could have led to transient shielding of the surface during glacial times, potentially distorting the cosmogenic isotope signal (Slosson et al., 2022). Yet our $^{10}$Be and $^{26}$Al-based denudation rates point to an integration time < 8,000 years, whereas for $^{14}$C, it is even shorter with < 3,000 years (Table S6). These ages are significantly younger than the last major glacial advance at around 12-11 ka ago (Ivy-Ochs et al., 2009). Therefore, we conclude that inheritance from pre-glacial surfaces does not significantly affect our data for all nuclides.





A further potential bias in quantifying denudation rates with cosmogenic nuclides could be introduced through sediment storage and reworking along the sediment cascade (e.g., Wittmann et al., 2020; Halsted et al., 2024). However, the ratios of the $^{10}$Be and $^{26}$Al concentrations are close to the values characterizing a nuclide production close to the surface (Fig. 6). Additionally, the resulting denudation rates are overall in good agreement with each other, at least in the upper zone and at the basin's outlet. This suggests that the $^{10}$Be and $^{26}$Al concentrations do not record the occurrence of a significant erosional

transience during the past kyrs, at least if the nuclide concentrations in the samples from the upper zone and the downstream end of the Gürbe basin are considered. We therefore consider, and this has already been noted in the previous section, that the signals preserved by the concentrations of in-situ $^{10}$Be and $^{26}$Al do record a pattern of erosion and sediment generation that has been stable at least during the erosional timescale of both isotopes, which are several thousand years. We acknowledge, that in the lower zone the supply of material through landsliding does result in a measurable discrepancy between the $^{10}$Be and

$^{26}$Al-based denudation rates (Figs. 3, 5 and 6), a pattern which is discussed in section 5.4. We also note that the denudation rate pattern of the short-lived $^{14}$C is distinctly different from that of $^{10}$Be and $^{26}$Al, which renders interpretations thereof more complex (see next section 5.3).

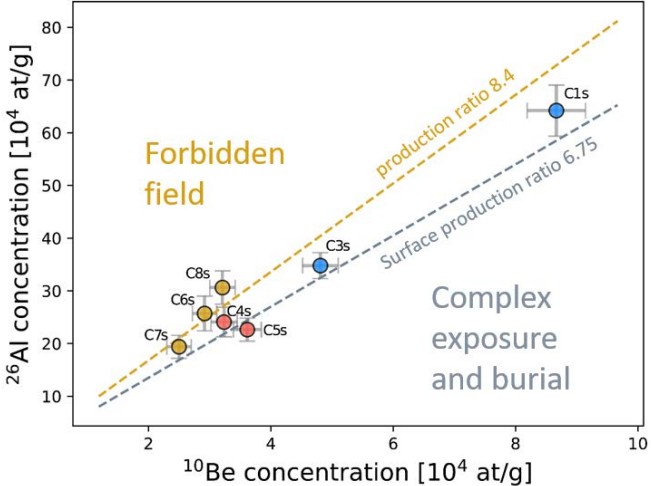

**Figure 6: Two-nuclide diagram showing the $^{10}$Be concentrations versus the $^{26}$Al concentrations.** The $^{26}$Al concentration is plotted against the $^{10}$Be concentration. The concentrations are within the field that is characteristic for a surface production. The occurrence of long-term burial can be excluded for our samples.

**5.3 Potential controls on the pattern of cosmogenic $^{14}$C**

$^{14}$C-based denudation rates have been shown to be sensitive to short-term perturbations (Hippe, 2017), particularly in

landscapes dominated by stochastic processes such as debris flows and landslides (e.g., Kober et al., 2012), or by storage of a significant volume of sediment (Hippe et al., 2012). However, we present three arguments why we are not capable of disclosing such controls on the $^{14}$C denudation pattern in our dataset. First, as outlined by Hippe (2017), in cases where the signal integration timespan exceeds the half-life of the corresponding cosmogenic isotope, a portion of the accumulated radionuclides



will decay before the material is completely eroded. The result is a reduction of the $^{14}$C concentrations, which could lead to an

artificially high erosion rate (Hippe, 2017). In our case, the minimum exposure time required to accumulate the observed $^{14}$C

concentrations is less than 3,000 years – well below the isotope's half-life of 5,730 years. This makes unaccounted radioactive

decay an unlikely primary cause of the low $^{14}$C concentrations in our samples. Second, a recent increase in erosion rates on

timescales shorter than the signal integration-time of $^{10}$Be and $^{26}$Al could explain why the $^{14}$C-based denudation rates are up to

three times higher than the estimates derived from the other isotopes. This is equivalent to a situation where the $^{14}$C signal is

already recording this increase in denudation, while the concentrations of $^{10}$Be and $^{26}$Al have not yet registered such a change.

While this could account for why all $^{14}$C-based denudation rates are higher than those derived from the other two nuclides, it

does not explain the relative differences in $^{14}$C-based denudation rates between the sampled catchments (Figs. 3 and 4).

Particularly, the pattern is not consistent throughout any of the three zones and the denudation rate variation within the zones

is higher than the variation between zones (Fig. 3). Third, geomorphic processes such as soil mixing or intermittent sediment

storage during transport could in part explain the observed differences between the $^{14}$C and $^{10}$Be/$^{26}$Al-derived denudation rates.

In particular, sediment could be stored – e.g., in response to landsliding – below the production zone of $^{14}$C, during which the

$^{14}$C inventory is partially lost due to the radioactive decay of this isotope (Hippe et al., 2012; Kober et al., 2012; Hippe, 2017;

Skov et al., 2019), a process which has been referred to as transient shielding by Slosson et al. (2022). Because the $^{10}$Be/$^{14}$C

ratios in the Altiplano samples range between 3.6 and 15.2 – significantly higher than the surface production ratio of 0.31 –

Hippe et al. (2012) interpreted the relatively low $^{14}$C concentrations in the riverine sediments as a record of storage rather than

surface erosion. However, since the $^{10}$Be/$^{14}$C ratios of the riverine samples in the Gürbe basin range from 0.4 to 0.75 and thus

do not fully fall within the complex exposure field, we consider it unlikely that the $^{14}$C concentrations primarily reflect a signal

of sedimentary storage. Furthermore, in the upper zone of the Gürbe basin, we exclude the occurrence of widespread storage

following sediment mobilization, as there is no evidence in the landscape that such processes have taken place (e.g., the

presence of large sedimentary bodies along the channels or talus cones as was reported by Slosson et al. (2022) from their

study area. In summary, while $^{14}$C has the potential to provide valuable insights into shifts in erosional dynamics, sediment

storage, and episodic mobilization of material from deeper levels (Hippe et al., 2012; Hippe, 2017), we are unable to

conclusively determine the specific processes responsible for the observed pattern of $^{14}$C concentrations and denudation rates

in the Gürbe basin.

**5.4 Landscape architecture and corresponding $^{10}$Be and $^{26}$Al signals**

The comparison of the geomorphic map with the $^{10}$Be and $^{26}$Al concentrations and the calculated denudation rates exhibits a

distinct difference between the upper and lower zone, each characterised by specific topographic features and dominant

erosional processes. Specifically, the landscape in the upper zone of the Gürbe catchment is characterised by smooth slopes

and the occurrence of partly incised channels with low steepness values and a low connectivity to the hillslopes. Such properties

are characteristic for a landscape where overland flow erosion, or hillslope diffusion according to (Tucker and Slingerland,

1997), have controlled the generation of clastic sedimentary material (van den Berg et al., 2012). The high cosmogenic nuclide




concentrations and, as consequence, the low denudation rates together with the $^{26}Al/^{10}Be$ concentration ratios of $7.41 \pm 0.69$ and $7.23 \pm 0.67$ – that are close to the surface production ratio of 6.75 (Balco et al., 2008; Nishiizumi et al., 1989) – are consistent with such an interpretation (Figs. 6 and 7). The similarity in the denudation rates calculated for the two long-lived

nuclides supports the interpretation of a stable undisturbed erosional regime without significant perturbations. We note that shallow landslides and localised rockfall do occur in this upper zone, but the resulting deposits are partly disconnected from the channel network, thereby minimizing their impact on the sediment budget of the Gürbe catchment (Fig. 2). In contrast, the lower zone exhibits a more dynamic erosional regime, where steeper slopes (20° to 25°) together with the predominant occurrence of mudstones in the Flysch and Upper Marine Molasse bedrock (Diem, 1986) offer ideal conditions for the

displacement of mid- to deep-seated landslides. Mapping also shows that landslides have impacted the sediment budget of the Gürbe River either through direct supply of sediment into the main channel, where it is then remobilised and transported downstream, or through erosion of the landslide bodies by tributaries, thereby reworking the previously displaced material. The stochastic nature of landsliding results in episodic sediment inputs with relatively low $^{10}Be$ concentrations, as material is exhumed from greater depths (e.g., Niemi et al., 2005). We use these mechanisms to explain the relatively high $^{26}Al/^{10}Be$ ratios

between $7.76 \pm 1.07$ to $9.54 \pm 1.17$ (Fig. 6) that we determined for the riverine material in three tributaries with material sources in these landslides. Additionally, the discrepancy between the $^{10}Be$- and $^{26}Al$-based erosion rates – with ratios ranging from 1.1 to 1.4 (Fig. 7) – suggests a complex erosional history involving material sourced from varying depths and subjected to different pathways during transport. These findings reinforce the view where the occurrence of landsliding not only results in an overall increase of erosion rates but also introduces a variability in the cosmogenic nuclide signals as seen here by the

$^{26}Al/^{10}Be$ concentration ratios.

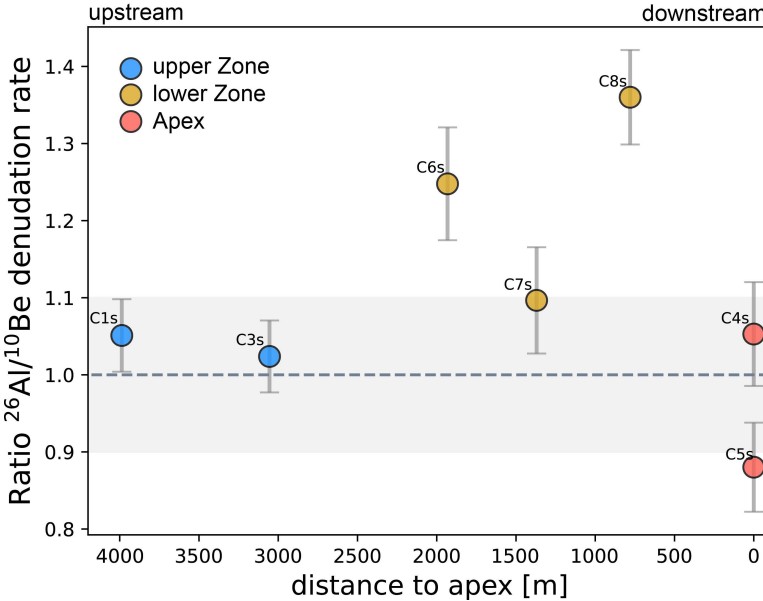



**Figure 7: Ratios between [26]Al- and [10]Be-based denudation rates.** In the upper zone the ratios between the denudation rate calculated with both isotopes are in very good agreement**.** In the lower zone the [26]Al-based denudation rates are lower than the [10]Be-based rate, leading to ratios that are larger than one. At the fan apex the ratios between both rates are in a relatively good agreement. The grey shading is representing the ± 10% envelope of the isotope ratio.

## 5.5 The importance of glacial conditioning as an erosional driving force

In a previous study, Delunel et al. (2020) showed that large part of catchment-averaged denudation rates in the European Alps can be understood as a diffusion-type of process, or a mechanism referred to as overland flow erosion (e.g., Battista et al., 2020) where denudation rates linearly increase with mean-basin hillslope angles until a threshold hillslope angle of 25°-30° (Schlunegger and Norton, 2013). Because the hillslope angles in the Gürbe catchment are below the 25° threshold, one would predict an erosional signal that could indeed be explained by such a mechanism. While indeed all our denudation rates fall in the predicted denudation rate range (cf. Fig. 5 of Delunel et al., 2020), and such an assignment is certainly correct for the upper part of the Gürbe catchment (section 5.4), the lower zone challenges this simplistic categorization. Indeed, despite the slopes being flatter than 25°, erosion in this region is dominated by deep-seated landslides that are perched by a channel network in which the eroded landslide material has been evacuated downstream. Such a combination of processes has typically been reported for landscapes steeper than 30° in the European Alps (Delunel et al., 2020) and other mountain ranges, e.g. the San Gabriel Mountains (DiBiase et al., 2023). This difference of erosion highlights the importance of another driving force than lithology alone because Flysch bedrock also occurs in the upper zone where landslides are largely absent, but where the hillslopes are less steep. Because the elevation of the knickzone separating the upper from the lower zones corresponds to the level of the LGM Aare glacier in this region (Bini et al., 2009), it is possible that the formation of the knickzone along the Gürbe River was conditioned by glacial carving during the LGM and possibly previous glaciations. Accordingly, the relatively flat channel and hillslope morphology in the area upstream of the knickzone could be explained by damming effects related to the presence of the Aare glacier. In contrast, the shape of the topography downstream of the knickzone was most likely conditioned by erosion along the lateral margin of the Aare glacier. Accordingly, the formation of a relatively steep flank could explain why the course of the Gürbe River is steeper than would be expected for a river with a same catchment size (do Prado et al., 2024), and why erosion on the hillslopes has been dominated by landsliding instead of overland flow erosion. Similar controls were already proposed by Norton et al. (2010) and van den Berg et al. (2012) upon explaining the occurrence of landscapes dominated by gentle hillslopes upstream of glacially conditioned knickzones and steeper hillslopes downstream of them. In such a context, a lowering of the base level after the retreat of the LGM glacier would have initiated a wave of headward erosion, with the consequence that the entire catchment is in a transient stage of landscape evolution (Abbühl et al., 2011; Vanacker et al., 2015). Accordingly, while the Gurnigel Flysch and Lower Marine Molasse bedrock seem to promote the occurrence deep-seated landslides – thus overriding the expected diffusion-controlled behaviour typically observed in catchments with similar slopes elsewhere – a steeping of the landscape due to glacial carving needs also be considered as an additional condition accelerating erosion in the lower part of the basin.





## 5.6 Provenance tracing with concentrations of cosmogenic nuclides

This study demonstrates that concentrations of cosmogenic isotopes offer suitable information for allocating the source of sediments in drainage basins where other provenance tracing methods, in our case whole-rock geochemical compositions of the riverine material, yield non-conclusive results. The whole rock geochemical analysis discloses a picture that is characteristic for a basin where limestones and litharenites are the main lithological constituents, as already demonstrated by Glaus et al. (2019) and Da Silva Guimarães et al. (2021) for other basins in the Alps. Yet, in contrast to the aforementioned studies, the results of our study did not allow a differentiation of the erosional signals derived from the various parts of the Gürbe basin, because here the only difference in composition among the samples seems to relate to the relative proportion of limestone in the detrital material (section 4.4). However, the outcropping limestone units are only present locally in some of the headwater reaches (Fig. 2). This introduces a potential limitation: while our quartz-based cosmogenic nuclide measurements provide valuable erosion rate estimates, they are inherently blind to sediment generated from quartz-free lithologies, such as limestone. This raises the question of whether preferential erosion in limestone-rich areas could bias our cosmogenic signal, for example by diluting quartz contributions downstream. In principle, this could lead to an underestimation or distortion of denudation rates. However, in the case of the Gürbe basin, we consider this bias to be negligible for three reasons: (1) limestone units form prominent cliffs and appear more resistant to erosion than surrounding lithologies, (2) no large-scale alternation of quartz-bearing and quartz-free lithologies is observed in the field, and (3) the limestone outcrops are spatially limited and do not dominate catchment-wide erosion. Consequently, the lack of a geochemical erosion signal is consistent with the observation that limestone does not significantly contribute to the total sediment flux, supporting the representativeness of the cosmogenic nuclide-derived erosion rates – particularly when used to trace spatial variations in erosion mechanisms across the basin. Material tracing with concentrations of in-situ isotopes works in basins with spatially varying erosion rates and erosional mechanisms. In addition, because here landslides repeatedly supply sediments from deeper levels where the production ratio of $^{26}$Al and $^{10}$Be is higher than on the surface (Akçar et al., 2017; Dingle et al., 2018; Knudsen et al., 2020), the detrital material has a high $^{26}$Al/$^{10}$Be-concentration ratio. In contrast the ratio of the same isotopes in sediments generated by overland flow erosion is lower. At the basin outlet, the detrital material reflects the integrated effect of sediment generation by overland flow erosion, landsliding, and other processes, resulting in a cosmogenic isotope signal whose strength depends on the relative contribution of each erosional process within the upstream catchment. Here, we demonstrate that this concept holds even in a basin where the spatial contrasts in measured erosion rates are relatively low. In such cases, a reconstruction of both the sediment sources and the associated erosional mechanisms requires a dense sampling strategy, as implemented in this work – a need that was already emphasised by Clapuyt et al. (2019) and Battista et al. (2020).



## Conclusions

This study demonstrates that cosmogenic nuclides are ideal tracers for identifying the origin of detrital material in basins with spatially varying erosion rates. They not only allow us to determine the region where most sediment has been generated, but also yield crucial information to reconstruct the mechanisms through which erosion has occurred (e.g., overland flow erosion versus landsliding), particularly when multiple isotopes are used. In the Gürbe basin, we were able to reconstruct such patterns and mechanisms using concentrations of in-situ $^{10}$Be and $^{26}$Al measured in riverine quartz. However, the concentrations of in-situ $^{14}$C, which we also measured in the same samples, yielded non-conclusive results. Also in the Gürbe basin, we found that the erosional processes were different in the upper zone above the LGM trim line where the landscape is flat and sediment generation has been dominated by overland flow erosion, and in the region below it where the landscape is steeper and erosion has been accomplished by landsliding and fluvial incision. This points to a legacy of the current erosional mechanisms stemming from the glaciations, where carving of the valley flanks by the LGM (Late Glacial Maximum) and earlier glaciers steepened the landscape, thereby promoting erosion below the LGM margin following the glaciers' retreat. This initiated a wave of headward retreat and the formation of an erosional front separating an upper part with low erosion rates (overland flow erosion) from a lower part where erosion occurs more rapidly and has mainly been accomplished by landsliding. In the Gürbe basin, we identified this erosional front through the occurrence of a distinct knickzone in the longitudinal stream profile. This phase of headward retreat thus suggests that the Gürbe basin has been in a long-term transient state of topographic development where the current accelerated erosion and landsliding in the lower zone has been glacially conditioned. Yet despite this transient state of basin development, we find – based on the concentrations of cosmogenic $^{10}$Be and $^{26}$Al in the riverine material – that the pattern and rate of sediment generation has been quite stable and thus steady during the past thousands of years. This was surprising because landslides have the potential to introduce a stochasticity in the way of how erosion and sediment generation occurs. It thus appears that the transient adjustment of the Gürbe basin to post-glacial conditions has occurred in a near-steady, possibly self-organised way, resulting in sediment generation, which has occurred at nearly constant rates at least during the past thousands of years. Because cosmogenic $^{26}$Al and $^{10}$Be integrate erosional signals over millennia in the Gürbe basin—including periods during which environmental conditions have changed—we expect a similar pattern of sediment generation and transfer in the near future, even under the current warming climate. Consequently, the local authorities are likely to be confronted with the same sediment transfer mechanisms through the cascade of check dams as they are today.



**Author contributions**

CS designed the study, conducted the analyses and wrote the paper with support by FS, DM and BM. NA offered scientific advise during sample collection and preparation, and during the analysis of the AMS results. MC, CV, PG supervised the AMS

measurements of 26Al and 10Be and offered support for the interpretation of the MS data. NH performed the in-situ $^{14}$C extraction and measured the 14C concentrations. All authors contributed to the scientific processing and discussion of the results and to the drafting of the paper.

**Competing interests**

The authors declare that they have no conflict of interest

**Data availability**

All data used in this study are provided in tables or included in the Supplementary Materials. The swisstopo digital elevation models (DEMs) are publicly available.

**Acknowledgements**

We thank Julijana Gajic for training and supervising the lab work at the IFG as well as Priska Bähler for the total Al

measurements. This work was funded through the Trebridge project funded by the Swiss National Science Foundation (SNSF project No 205912).



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
