# Peer review of "Quantifying erosion in a pre-Alpine catchment at high resolution with concentrations of cosmogenic $^{10}\text{Be}$ , $^{26}\text{Al}$ , and $^{14}\text{C}$"

_EGUsphere, 2025_

## Author Response (AR1)

Response to reviewer 1:

We thank the reviewer for the careful evaluation of our manuscript and the constructive comments. The suggestions have been very helpful for us to improve the manuscript regarding the clarity and overall quality. We carefully revised the manuscript to address all points raised.

Upon revising the manuscript, we mainly addressed the following points:

- Explaining in more detail the character of the landslides within the catchments and incorporate this in more detail in the discussion section 5.4
- Clarifying the points raised in the methodology section
- Correcting typographical errors throughout the text
- Rectifying minor oversights, such as reporting 26Al concentrations with 1 Sigma (instead of 2 Sigma) uncertainties in the text, or consistent rounding to the same digit.

Below we provide a detailed, point-by-point response to each comment.

Reviewers Comment: The authors state that the Gürbe catchment is large enough to ensure good sediment mixing despite the presence of deep-seated landslides. From the background, however, it seems that these landslides are slow and episodic, only supplying minor amounts of sediments at a time. Are they truly comparable, in terms of providing shielded sediment, to a single large deep-seated landslide event? I felt the background information on landslides was insufficient to fully assess this distinction.

We agree with the reviewer that the landslide mechanism could have been explained in more detail. We addressed this by including a paragraph where we provided more details about the character of the landslides within the catchment. We also expanded the text in section 5.4 where we discuss the related implications.

Reviewers Comment: L30: Typo. "aggregated"

Corrected.

Reviewers Comment: L93: How long are the stable phases? Please relate them to the nuclide integration timescales.

We expanded the explanation about the timescale of the landslide recurrence time. We took the opportunity to provide a more detailed description of the landslide processes. The relation to the nuclide integration time was added in the discussion section 5.4.

Reviewers Comment: L131: Please, indicate, which resampling method was used (bilinear, bicubic nearest, etc.).

We specified the resampling method.

Reviewers Comment: L134-5: Please rephrase the sentence about the DEM correction — it is currently difficult to follow.

We revised the sentence for more clarity.

Reviewers Comment: L142: It would be helpful to include the equation for the connectivity index.

We added a brief explanation on the functionality of the Index and we referred to the original paper (Cavalli et al. 2013) where the related equations are explained. We did so because we merely used the toolbox and did not modify any equations or parametrizations.

Reviewers Comment: L151: There are many ways to calculate ruggedness/roughness using different approaches and kernel sizes. Please add a brief explanation of how ruggedness is defined here.

We added a brief explanation on the terrain ruggedness index elaborated by Riley et al. (1999), which is the approach we used in our work.

Reviewers Comment: L152: as any smoothing of stream elevations used for Ksn calculation? Or were raw elevations used for each pixel KSn, later averaged within 125 m of the stream?

We used the elevations from the DEM for each stream pixel and then averaged them along the specified stream segment length (i.e., the default TopoToolbox2 implementation). We clarified this in the text.

Reviewers Comment: L161: Diffusive hillslope transport is negligible? Seems surprising. Later in the discussion the manuscript talks about diffusive hillslope transport in the upper zone.

This seems to be a miscommunication, which we clarified in the revised version.

Reviewers Comment: L199: Figure 1 legend says sampling occurred in 2017 and 2022; the caption says 2017 and 2020; the text says 2022 and 2023. Please clarify.

We apologize for this oversight, and we corrected the text accordingly.

Reviewers Comment: L213: How far above the stream was this sample taken? Was it from a recent (hundreds of years) deposit, or from something older, e.g., Pleistocene?

We clarified this in the revised manuscript.

Reviewers Comment: L271: Typo. "Influences"

Corrected.

Reviewers Comment: Section 4.1: As a reader, I would appreciate some representative photos of the different zones to better judge the interpretations made. Especially, when it comes to the erosion process discussion.

We thank the reviewer for this suggestion. We added a new figure with representative photos in the revised manuscript.

Reviewers Comment: L321: Since production rates increase with altitude, please put your reported concentration differences into the context of corresponding production rate differences between zones.

Our aim at this position in the text is to describe the nuclide concentration pattern without any interpretation. We note here that the differences in concentrations are much larger than the differences in average production rates between the different zones. However, comparing production rates at this point is not straightforward due to their dependence on the scaling method, slight differences in production pathways, snow shielding, and other factors. For this reason, Cronuscalc v3 no longer reports scaled production rates (Balco, 2020). Therefore, adding quantitative production rate differences would require adding much text (also in the method section) that would yield similar ratios as the calculated denudation rates after adjusting for altitude. We do not consider larger changes (>10m) in altitude to have occurred during the signal integration period. Therefore, instead of reporting average production rates, we refer to the denudation rate results, which already account for the difference in production and consider all the mentioned factors. We adapted the respective sentences to clarify this point.

Balco, G.: The bleeding edge of cosmogenic-nuclide geochemestry: Where is the production rate?, https://cosmognosis.wordpress.com/2020/06/10/where-is-the-production-rate/, 2020.

Reviewers Comment: Table 1: Add a column indicating the zone of each sample to facilitate interpretation.

We added this column in table 1.

Reviewers Comment: Figure 3: Consider plotting the 10Be–26Al and 10Be–14C ratios, either here or elsewhere, to support later discussion.

In Figure 6, we plotted the 26Al concentration against the 10Be concentration, thereby displaying the ratios in relation to the surface production ratio. We therefore added a new figure (8) showing the same type of plot, with 10Be against 14C.

Reviewers Comment: Figure 4: the basin outline color is hard to interpret. I assume this was done to avoid overlapping colors in nested catchments, but the outlines may need a much thicker rim. Since you used TopoToolbox, here's a useful blog post on creating thicker outlines:

https://topotoolbox.wordpress.com/2021/12/17/making-beautiful-drainage-basin-outlines/

We agree that visualising the data for nested catchments is a challenging task. We adjusted the outlines to facilitate easier interpretation.

Reviewers Comment: **Section 5.3 – 14C interpretation:**

L366: The 14C pattern appears relatively similar to the other nuclides, with some offset and one outlier in the middle section. In the upper section, it shows a similar difference between the two samples compared to the other two nuclides. This statement makes it look like the 14C is all over the place.

Other studies have reported an increase in erosion over the past 3000 years in the Alps (Andrič et al., 2020; Rapuc et al., 2024), which matches the integration time-span of 14C in this study. Given its sensitivity, 14C is more prone to variability from stochastic processes and shielding, especially with a small sample set. Personally, I would lean toward one of the authors' proposed explanations for the 14C results, but I understand and respect their conservative interpretation.

We thank the reviewer for adding depth to this discussion. However, we note that the variability in the 14C concentrations (and denudation rates) within one sample is much larger than the variability between the different samples.  Furthermore, as discussed in section 5.3, the observed pattern does not align well with proposed interpretation, nor with different interpretations of other 14C inventories in other studies. This leaves us with little room for other interpretations for our results. However, we expanded the discussion of a potential increase in erosion rate as cause for the 14C pattern to communicate more clearly the observations that led to this interpretation.

Reviewers Comment: L490. Typo. Adjust parentheses of citation.

Corrected.

Reviewers Comment: Section 5.4 Please provide more specific (and, if possible, quantitative) discussion of the mechanisms affecting the low-reach samples. How exactly would landslides and related cascading processes influence the 10Be/26Al ratios? Some simple calculations could strengthen this section. For example, using your mapping and published information on landslide depths and sediment volumes, can you estimate how much landsliding or storage over what timescales would be required to explain the measured concentrations?

We agree that this would be an interesting avenue for further research. Yet, we consider such a quantification beyond the scope of this study as it would require detailed modelling and a detailed knowledge about the sediment fluxes in space and time, which we don't have at the required resolution level, unfortunately. We therefore included further sentences to explain the landslide advection mechanism and the resulting sediment cascade together with some more specific qualitative explanations.

Reviewers Comment: Figure 7 shows that 10Be/26Al ratios in the fan samples roughly equal surface values of the upper catchment zone. If erosion rates are higher in the middle section, and given its larger quartz-bearing area, one would expect the fan deposits to be dominated by that flux. Yet the ratios are lower in the fan than in the middle section. This discrepancy is not addressed — what do the authors think about this?

The figure 7 (now 9) displays the good agreement between the 10Be and 26Al-based denudation rates, not the nuclide concentration ratio. We highlight here that for some sub-catchments the 10Be-based rates are higher than the 26Al-based rates (i.e., C6s, C8s) in the lower zone, while for other they are not, which also includes C7s in the lower zone. The higher 10Be-based rates suggest a relative deficit of 10Be compared to 26Al (already adjusted for production rate differences; see related comment above). There are several possibilities why we do not record a similar 26Al surplus/10Be deficit in the downstream samples at the apex:

1) The effective sediment mass input from C6s and C8s is comparatively small, and, therefore, the signal is diluted rapidly when mixed with sediment from overland flow erosion from higher in the catchment. This overland flow sediment has also in general higher nuclide concentrations due to longer residence time at the surface at higher elevations (see Fig. 7 new). Assuming complete mixing, a mass flux ratio of 2:1 (upper to lower) could already completely eradicate the nuclide imbalance (i.e., balancing ratios of 8.8 [C6s] and 7.4 [C1s] to ~7.9 which would align well with the 7.4 +/- 1.0 ratio of C4s).

2) The nuclide imbalance is a reflection of stochastic landslide output that is locally fluctuating over short timescales. Hence, if mixing is assumed, the material in the apex samples that originated in C6s, C8s has a different ratio than the material currently delivered by the sub-catchments. A potential hint for this could be the fact that C7s is very different from C6s, C8s, and that C4s and C5s are internally different (in both ratios and denudation rates).

3) The mixing is incomplete. The material with the Al surplus might be intermediately stored in the channel network close to the surface. With time and nuclide production at the surface production ratio, the imbalance will decrease over time.

In reality, most likely all three effects contribute to differences in ratios. However, currently, without constraints on the relative mass fluxes we do not have the information needed to test or quantify these effects. We added a brief statement that convey these notions in section 5.3.

Reviewers Comment: L557: Please clarify why limestone would lead to underestimation or distortion of denudation rates. Do you mean that rapid limestone erosion would not register in quartz-based estimates? This seems obvious, so I found the list in lines 560–565 somewhat distracting. Perhaps streamline it.

We clarified this point. In particular, the total sediment flux might be different as the 10Be-based denudation rates are blind to limestone lithologies. Hence, the 10Be or 26Al-based denudation rates would not be affected but not be a good proxy for the actual surface denudation rate. We reformulated the text accordingly.

Reviewers Comment: L595: The time-scale of knickpoint migration and the integration time of the cosmogenic nuclides are very different, which is something that should be highlighted in this statement or elsewhere. With the measured erosion rates the authors could even think about calculating knickpoint propagation times.

We agree on this point. We considered quantifying the propagation rates since the termination of the LGM. However, we lack geomorphic constraints to reconstruct in detail the bed morphology of the LGM glacier. Therefore, we are not able to exactly determine the location where the knickpoint started to retreat headward. Accordingly, we cannot determine the knickzone retreat rates.

Response to Reviewer 2:

We thank the reviewer for taking the time to carefully read and evaluate our manuscript. We appreciate the positive and encouraging feedback. In response to the minor suggestions provided by Reviewer 1, we have made corresponding adjustments in the final version to address these comments and further improve the manuscript. Details of the revisions can be found in our detailed response to Reviewer 1.